# Just ten percent of the global terrestrial protected area network is structurally connected via intact land

Michelle Ward ⬤ [1,2✉], Santiago Saura[3,4], Brooke Williams ⬤ [1,2], Juan Pablo Ramírez-Delgado ⬤ [5], Nur Arafeh-Dalmau ⬤ [1,2], James R. Allan[2,6], Oscar Venter[5], Grégoire Dubois ⬤ [3] & James E. M. Watson ⬤ [1,2,7]

Land free of direct anthropogenic disturbance is considered essential for achieving biodiversity conservation outcomes but is rapidly eroding. In response, many nations are increasing their protected area (PA) estates, but little consideration is given to the context of the surrounding landscape. This is despite the fact that structural connectivity between PAs is critical in a changing climate and mandated by international conservation targets. Using a high-resolution assessment of human pressure, we show that while ~40% of the terrestrial planet is intact, only 9.7% of Earth's terrestrial protected network can be considered structurally connected. On average, 11% of each country or territory's PA estate can be considered connected. As the global community commits to bolder action on abating biodiversity loss, placement of future PAs will be critical, as will an increased focus on landscape-scale habitat retention and restoration efforts to ensure those important areas set aside for conservation outcomes will remain (or become) connected.

[1] School of Earth and Environmental Sciences, University of Queensland, Brisbane, QLD 4072, Australia. [2] Centre for Biodiversity and Conservation Science, The University of Queensland, Brisbane, QLD 4072, Australia. [3] European Commission, Joint Research Centre (JRC), Via E. Fermi 2749, I-21027 Ispra, VA, Italy. [4] ETSI Montes, Forestal y del Medio Natural, Universidad Politécnica de Madrid, Ciudad Universitaria s/n, 28040 Madrid, Spain. [5] Natural Resources and Environmental Studies Institute, University of Northern British Columbia, Prince George, BC, Canada. [6] Institute for Biodiversity and Ecosystem Dynamics (IBED), University of Amsterdam, P.O. Box 942401090 GE Amsterdam, The Netherlands. [7] Wildlife Conservation Society, Global Conservation Program, Bronx, NY 20460, USA. ✉email: m.ward@uq.edu.au

Protected areas (PAs) are a core tool in abating the biodiversity crisis[1,2], and their importance is reflected in the 2020 Strategic Plan for Biodiversity[3]. This international agreement calls for the expansion of the global PA network to cover 17% of terrestrial areas and 10% of marine areas by 2020. Crucially, the Strategic Plan stipulates that PA networks must be well connected, effectively and equitably managed, and also cover ecologically representative areas of particular importance for biodiversity[3]. Nations are currently negotiating the Post-2020 Global Biodiversity Framework[4] and a net increase in PA connectivity is increasingly being considered a critical component of a number of biodiversity conservation goals[4]. For example, the new biodiversity strategy for 2030 of the European Union is not only setting a target of 30% of land and 30% of sea to be protected, but stresses the need of a coherent and resilient Trans-European Nature Network in which ecological corridors will be essential[5].

Due to their extraordinary importance for biodiversity outcomes, PAs have received substantial attention in global conservation policy discussions and research, with recent assessments focusing on how well they are representing species[6], their overall management effectiveness[7,8], and how well are they abating threatening processes[9,10]. Yet, to date, reporting is almost completely blind to how well connected the expanding global PA estate is, with only substantive research conducted at country and region scales[11–16], or solely considering connectivity through protected land[17,18], disregarding the condition of the wider landscape context. One way to assess how well PAs are connected is via measuring the structural connectivity across the landscape[19] through analyzing the quality and extent of surrounding habitat[20]. While structural connectivity alone does not guarantee connectivity for all species[21], high levels of landscape connectedness is seen as critical for species adaptation under anthropogenic climate change as it facilitates individuals and populations to track their preferred microclimates[22–24]. Under projected climate scenarios, it is predicted that many species will need to move further and more rapidly in the 21st century and connected landscapes that facilitate this movement is one of the best conservation responses[25,26].

Structurally connected landscapes allow fundamental ecological mechanisms to operate unimpeded, such as meta-population retention[27] and successful dispersal and migration[28,29]. Beyond species-specific benefits, structurally connected landscapes allow for increased ecosystem function and resilience[30] by ensuring nutrient cycling can continue unabated, as well as other important abiotic conditions, such as radiation, wind, light regimes, humidity, and key hydrological regimes[31,32]. It is well known that land uses such as farming, urbanization, mining, and unsustainable forestry disrupt the connectivity of landscapes to various degrees[28,33]. The retention, and where necessary, restoration of connectivity across a landscape matrix between PAs is therefore vital for achieving biodiversity goals outlined in the Strategic Plan for Biodiversity[3].

Here, we analyze the structural connectivity of the global terrestrial PA system using measures of both the probability that connectedness can be achieved and the contiguity of intact land (i.e. areas largely devoid of high anthropogenic pressures that significantly alter natural habitat). In this analysis, we assume species can move more freely between PAs through intact land[28]. We determine the structural connectivity of the global network of PAs by quantifying intact continuous pathways between PAs. To assess landscape intactness, we use the human footprint dataset (HFP), which, at a 1 km$^2$ resolution, is the most comprehensive, fine-scale human pressure map available as it takes into account agricultural lands, roads, railways, human population density, built environments, night-time lights, and navigable waterways,

all of which are driving the species extinction crisis[34]. We use the most up-to-date HFP layer (2013), which supersedes the published version dated 2009[35]. Following other studies[9,36], we define intact land with a HFP value <4 out of 50, as this threshold is where anthropogenic activities significantly change the state of land from largely natural in extent to highly modified[37]. Moreover, this human pressure threshold is associated with the sharpest declines in mammal movement[28], erodes behavioral diversity[38], and is one of the strongest predictors of mammal species extinction risk[39]. As functional connectivity and structural connectivity are positively correlated[19], we argue that areas with human pressure above this threshold are unlikely to hold sufficient connectivity value for many key elements of biodiversity. Using this HFP threshold, we find that only 9.7% of Earth's terrestrial protected network can be considered structurally connected via intact landscapes. However, recognizing that there is no one true level of pressure that prohibits connectivity for all biodiversity (as ecological responses to human pressure are idiosyncratic), we provide several sensitivity analyses around this HFP threshold, finding our results are robust to lower and higher HFP values (Supplementary Tables 1, 2). This finding highlights the need for not only better placement of future protected areas and other area-based conservation efforts but also for more bolder habitat retention and restoration goals to ensure wider landscape and regional scale connectivity outcomes are achieved.

## Results

**How structurally connected is the global protected area network?** Globally, while 41.6% of terrestrial land is intact, only 9.7% of the area under protection can be considered structurally connected through intact landscapes (Fig. 1 and Supplementary Fig. 4). This means that very few PAs have a fully continuous pathway through intact lands, connecting their demarcated edges. At a continental scale, PAs in Oceania are the most connected across all continents (16.8%), followed by the Americas (14.8%). In contrast, Asia (3.2%), Africa (0.5%), and Europe (0.3%) have extremely low levels of PA connectivity provided by intact lands.

**National scale reporting**. At a national scale, the percentage of structurally connected PAs varies enormously (Fig. 2a and Supplementary Figs. 5–8). The majority of countries and territories maintain the lowest level of structural connectivity possible (median connectivity = 0%). These countries and territories are found not just concentrated in Europe (where one would expect minimal presence of low human pressure matrix between PAs) but also across Asia and Africa (Supplementary Data 1) where landscapes are rapidly changing through large-scale infrastructure projects such as roads and agriculture[40,41]. This has significant ramifications for international conservation agendas, as many Asian and African countries and territories are megadiverse when it comes to biodiversity[42,43]. For example, Vietnam—one of Earth's most biologically diverse countries[44]—has ~8% protection and no connected PAs based on our analysis (Fig. 2b). Nevertheless, PA connectivity is likely vital for the persistence of critically endangered species such as saola (*Pseudoryx nghetinhensis*) and Indochinese tiger (*Panthera tigris corbetti*)[45,46]. Similarly, Madagascar is home to some of the most genetically-diverse species on Earth, including the black and white ruffed lemur (*Varecia variegata*), aye-aye (*Daubentonia madagascariensis*), and fossa (*Crytoprocta forex*). These genetically unique species, being predominantly arboreal, require contiguous intact landscapes to fulfill their important ecological roles[47], yet we found that Madagascar only has 4.2% intact land remaining and no fully connected PAs.

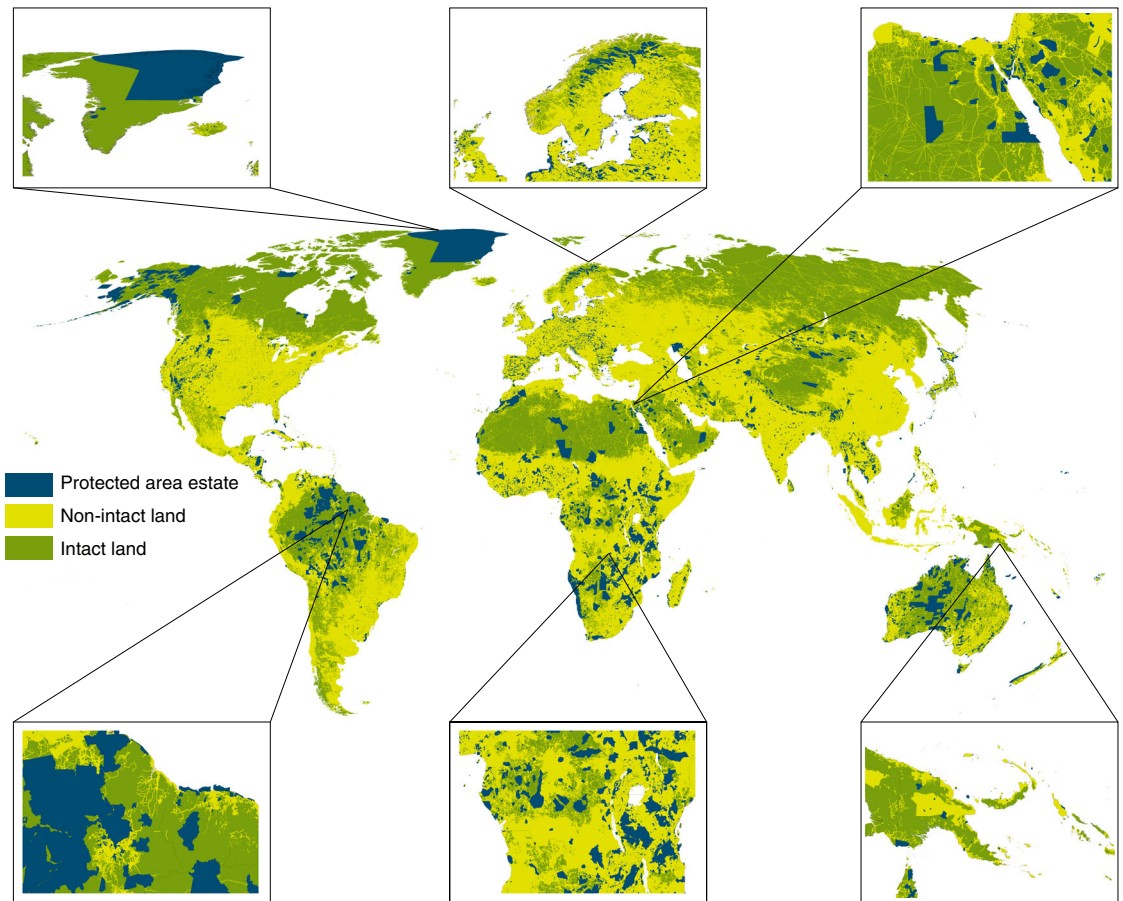

**Fig. 1 Human pressure compromises structural connectivity of protected areas.** The spatial distribution of the PA estate (blue) littered throughout nonintact (yellow) and intact land (green), as ascertained using the HFP (2013). Intact land was defined as having a HFP < 4 (following Beyer et al.[36]). We provide six fine-scale examples, starting top left and moving clockwise, Greenland, Finland, Egypt, Papua New Guinea, Democratic Republic of Congo, and Guyana.

Only nine (4.6%) countries and territories have >17% of their land protected (which many countries and territories define as their contributions to the CBD PA target)[48] and maintain >50% structurally connected across their PA network (Fig. 3). These countries and territories with high proportions of land under protection have statistically significant correlations with higher proportions of connected PAs ($\rho = 0.28$, $n = 183$, $p < 0.05$). We found no significant relationship between countries or territories with high proportions of the protected land connected and the number of PAs ($\rho = 0.00$, $n = 183$, $p > 0.05$) or the size of the country or territory ($\rho = 0.04$, $n = 183$, $p > 0.05$). This indicates that structural connectivity may not be considered when countries and territories are adding to their PA estates. For the more intact countries and territories, this may be because they declare fewer but larger PAs, while more human-modified countries and territories (such as those in Europe) declare many small PAs that are usually surrounded by a nonintact matrix. It may also be the result of PAs planning and management operating under multiple jurisdictions (e.g. federal, provincial/state, or municipal) within one country or territory[49,50], with a lack of structural connectivity of PAs established occurring simply because of a lack of coordination between jurisdictions.

Eighteen countries and territories have >50% of their land that can be considered intact, yet have very low structural connectivity between their PAs (Fig. 4 and Supplementary Figs. 9, 10). For example, our analysis revealed that Egypt has 77.1% intact land, but only 10.8% of its PA network is connected. Even though there are large PAs within this nation, including El-Gelf El-Keber

(48,523 km$^2$) and Elba (36,600 km$^2$), major roads, agriculture, and urban sprawl are increasingly fragmenting the landscape[51].

The vast majority (76.4%) of countries and territories have few intact landscapes remaining and low proportions of connected PAs. These include surprising examples such as the Democratic Republic of Congo (DRC), which we found to have 23.3% intact land and no connected PAs. Much of the PA network within the DRC is scattered across the country, disconnected by agricultural lands, extractive industries, and roads[52]. In addition to the 149 countries and territories that have low proportions of both intact landscapes and few connected PAs, there are also 13 countries and territories that do not have any remaining intact landscapes, and therefore, no structural connection between PAs. Statistically, countries and territories with low proportions of intact land also have significantly lower connected PAs (Spearman's $p = 0.40$, $n = 183$, $p < 0.05$; Eigenvalue 1.37, Eigenvalue 0.62).

**Discussion**

Our results show that while some countries and territories are meeting the areal component of global PA targets, much of this PA estate is not connected due to anthropogenically-modified habitat. This suggests that the overall strategic goal of preventing further biodiversity loss will be likely compromised without increased focus on wider land-use efforts to retain and restore natural habitats beyond PA boundaries. Any plan for maintaining and restoring ecosystem connectivity between PAs must include a clear, quantifiable focus on retention strategies for remaining ecosystems that are currently not degraded, because these places

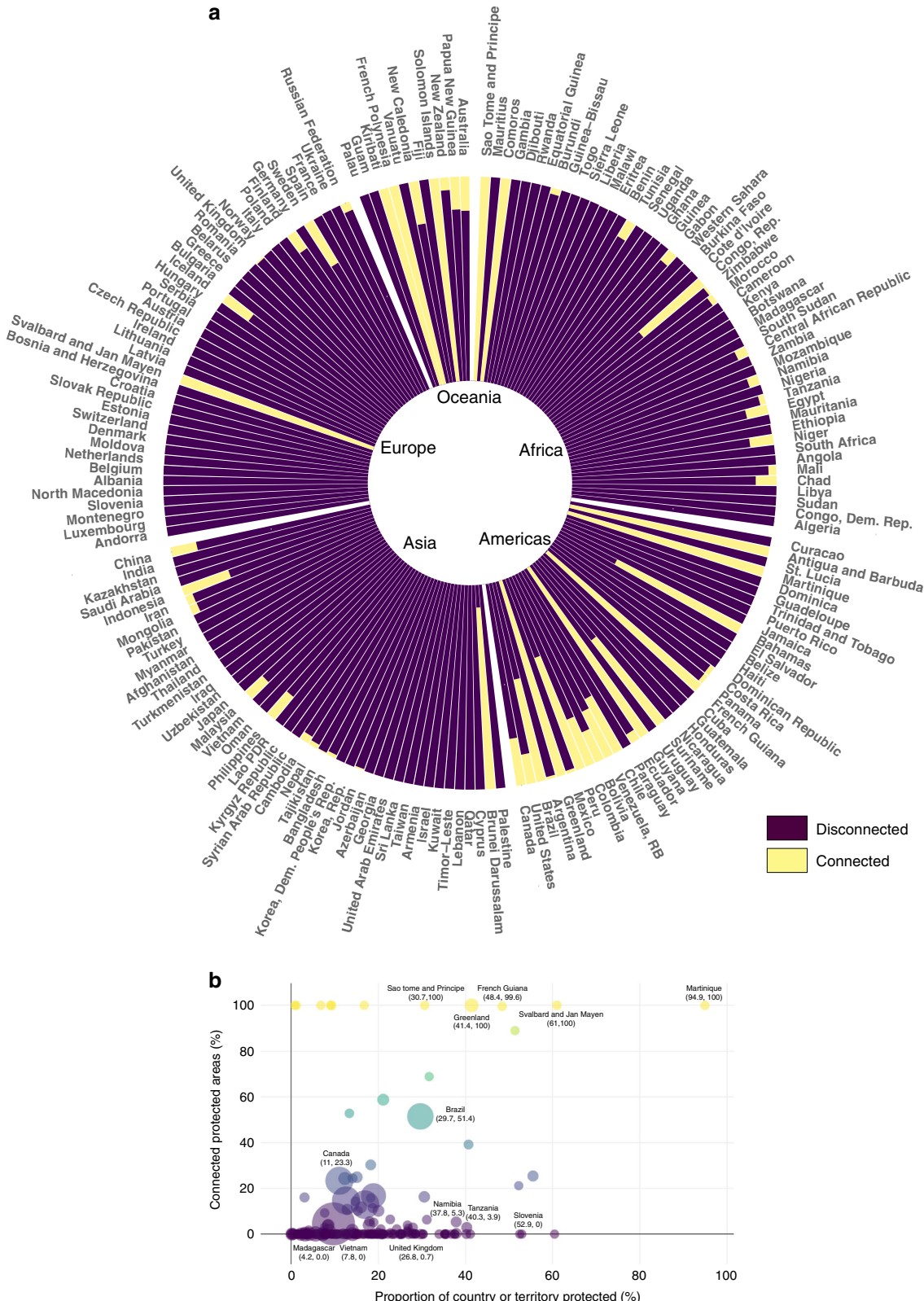

**Fig. 2 Few countries and territories maintain a structurally connected PA network. a** Proportion of connected and disconnected PAs for each nation. Countries and territories are grouped per continent and are ordered based on smallest to largest landmass (moving clockwise). **b** Relationship between the proportion of land under protection (*x*-axis) and the proportion of connected PAs (*y*-axis) per country and territory (less connected countries and territories are purple while more connected countries and territories are yellow). The size of the bubble indicates the size of the country and territory.

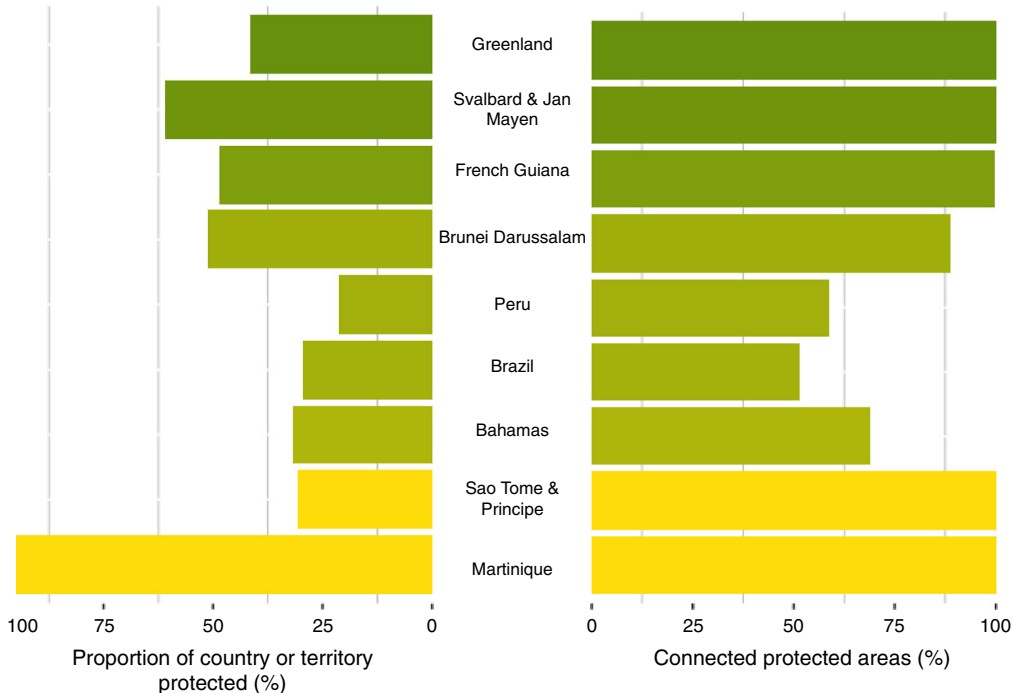

**Fig. 3 Examples of high proportions of land protected and structurally connected PAs.** Only nine countries and territories protect >17% of land (left) and maintain >50% connected PAs (right). Countries and territories are ordered based on the highest proportion of remaining intact land (less intact countries and territories are yellow, while more intact countries and territories are green).

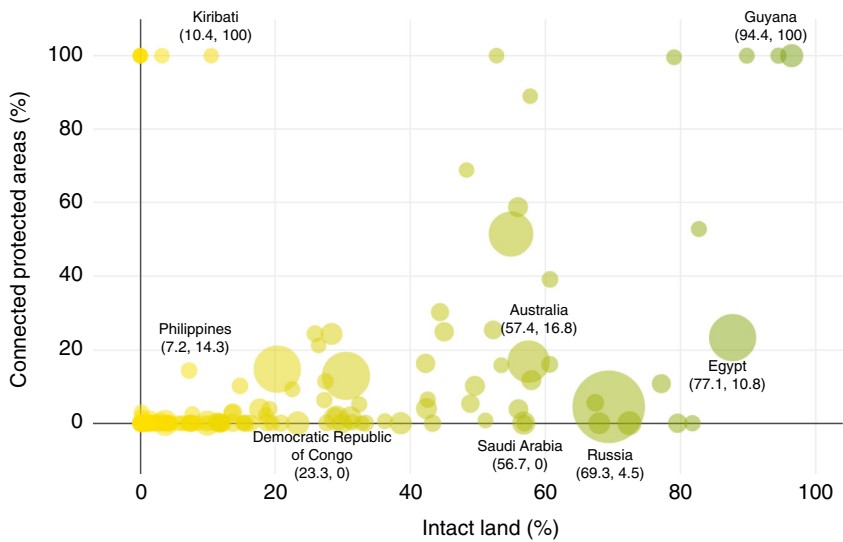

**Fig. 4 Relationship between intact land and structurally connected PAs.** Scatterplot showing the relationship between proportion of intact land (less intact countries and territories are yellow, while more intact countries and territories are green) and proportion of connected PAs per country or territory. The size of the bubble indicates the size of the country or territory.

are, by definition, areas maintaining high connectivity and known to hold exceptional value[28,33]. It is also well accepted that avoiding degradation of habitat (and hence loss of connectivity) is a far better strategy than attempting restoration after it is lost. This is because restoration is more costly, riskier, and unlikely to lead to the full recovery of structurally connected values[53].

For Earth's remaining intact and connected areas to be retained, they must be formally recognized, socially accepted, prioritized in spatial plans, economically viable, and then effectively managed, so they can be protected from human impacts[54].

Taking a coordinated approach of all factors can greatly benefit long-term connectivity as it provides the opportunity to identify areas that are most at risk of alienation, acquire patches of key importance to maintain connectivity, and can provide economic, social, and cultural needs for people[55]. These intact and connected areas are expanding beyond strict PAs. For example, a definition for 'other effective area-based conservation measures' (OECMs) was agreed by nations in November 2018, with the objective to achieve positive and sustained long-term outcomes for the in situ conservation of biodiversity, with associated

ecosystem functions and services and where applicable, cultural, spiritual, socio–economic, and other locally relevant values[56]. The global extent of identification and reporting of OECMs is expected to increase rapidly over the coming years[57]. As such, their use, if planned and implemented well, could play an enormous role in keeping PAs connected.

But our findings show that even if humanity was successful in halting the degradation of all remaining intact ecosystems, there would still be many isolated PAs given their surrounding matrix has already been highly altered. As such, there is also the need for a broad, restoration agenda to rapidly increase structural connectivity between areas set aside for conservation. There is clearly appetite for such bold restoration action, with the UN recently declaring the 'Decade on Ecosystem Restoration'[58], by which 350 million hectares of degraded land will be restored between now and 2030. We argue that these types of restoration goals should be framed within a broader connectivity agenda and specifically planned to maximize the quality of the landscape matrix between PAs, as well as degraded land inside PAs essential to biodiversity outcomes. While we did not consider the condition inside PAs, we know that approximately one third of land protected is under intense human pressure[9]. Restoration of degraded land inside PAs or less degraded natural ecosystems outside but near PAs presents some of the most cost-effective restoration opportunities[59], and lowest potential for conflicts with other priorities (such as agriculture). In addition, it is essential to incorporate the cost of such restoration and conservation actions into other societal goals.

Our results highlight the need for a far more comprehensive reporting framework on area-based conservation that captures not just the extent and overall effectiveness of local implementation of conservation activities, but also how connected these PA networks are within the wider landscape. As the relationship between the placement of area-based conservation activities, how they are managed, and the wider landscape context is nuanced, metrics on structural connectivity must be integrated with other assessments of PA effectiveness. For example, while Venezuela is currently achieving 55.5% protection, we found very little structural connectivity between its PAs. This may not be a poor result, as the PA estate within Venezuela seems to be large, well-managed, and representative[60]. Other countries and territories may have low proportional connectivity because their PAs have been established in human-dominated landscapes, which are often where much conservation action must occur. This type of reactive conservation strategy is clearly necessary for countries and territories that have degraded, but ecologically important landscapes[61]. In contrast, some countries and territories that have good connectivity scores could be hiding inherent biases in PA placement. For example, Australia is achieving ~17% structural connectivity via our assessment and has ~19% of its land under protection, which could be considered a relatively good outcome. Yet, upon closer inspection, most new PAs have been placed in the desert ecosystems, which do not adequately represent all taxonomic groups and are already well represented in the nation's PA estate[62]. In addition, most Australian PAs are also not effectively managed, with 1390 threatened species continuing to lose critical threatened species habitat inside and outside PAs[63,64]. The numbers around PA structural connectivity alone can hide important issues, but there are methodologies that capture quality, extent, and now connectivity of landscapes[36] and it should be possible for countries and territories to transform these into a framework that holistically assesses the overall effectiveness of area-based conservation action.

The method we use here is easily replicable. As such, it could provide a metric that governments can use, and report on, when creating new PAs that ensures this expansion results in a more connected PA network. The metric can also be used by the global community to measure, track, and implement global connectivity goals. For example, this effort to measure structural connectivity between PAs can be easily integrated with the policy implementation framework called the 'Three Conditions for Nature', that has been proposed to the CBD[65,66]. This framework identifies three broad conditions of terrestrial areas based on land-use drivers and pressures. The three conditions include cities and farms (~18% of Earth), shared lands (~56% of Earth), and large wild areas (~26% of Earth)[65]. Unsurprisingly, when structural connectivity between PAs is assessed under the Three Conditions for Nature framework, the majority of countries and territories maintaining high proportions of connected PAs predominately also had high levels of large wild areas (Supplementary Data 2). Similarly, when countries and territories have minimal connected PAs, the proportion of land within cities and farms and shared lands is predominately high. These areas require a very different set of actions varying from restoration to retention of ecological integrity. By including these types of structural connectivity assessments within implementation frameworks like the Three Conditions for Nature, it is possible for decision makers to generate the different policy targets (e.g. more restoration in shared landscapes and more protection activities in wild landscapes) around connectivity that may lead to better outcomes.

We note that our analysis is subject to some caveats. While the HFP is one of the most widely used and comprehensive datasets on global human pressure[28,39,67], it does not capture all human activities[35]. For example, the HFP does not include walls or fences which impose significant restrictions on species migrations and movement in some places[68]. Invasive species are also a major contributor to the degradation of entire ecosystems and reduce functional connectivity[69], yet they are not directly mapped by the HFP. As such, our results may present an over-estimation of connectivity. Second, our analyses do not include some PAs that are not included within the WDPA[70] and when these places are mapped and embedded in the WDPA, the analysis should be updated. We also note that structural connectivity is not the same as ecological connectivity, which has many components including species-specific functional connectivity. Functional connectivity measures the processes by which subpopulations of species are connected into a demographic unit[19], and can be evaluated through strict adjacency[71], threshold distances[72], or resistance-weighted functions[73]. While resistance-based approaches have also been used to evaluate structural connectivity between PAs[16] and connectivity of the PAs to the surrounding landscapes[11], we did not use a resistance-based approach here because we wanted to take advantage of the latest HFP and the evidence that has emerged around the thresholds chosen[28,36,39]. There is an important research gap to fill the links between our measures of structural connectivity to these efforts to measure functional connectivity.

We also note in some cases, a low proportional connectivity score between PAs may be a reasonable outcome. This is likely the case in some parts of Europe, where small-scale efforts to create corridors between PAs may not be captured at the 1 km$^2$ resolution of our analysis. In addition, areas of high pressure in the matrix between PAs may be acceptable for some species that have co-existed with human-modified landscapes for centuries. But as most species, especially those that are endangered, cannot persist in human-dominated landscapes[28,39], our results should provide a sobering assessment for many countries and territories.

The retention and restoration of intact landscapes that surround PAs is critical to abating the biodiversity crisis. Right now, the majority of PAs are isolated by a matrix of rapidly eroding intact habitat and are unlikely to be as effective as they could, especially when considering the likely consequences of

anthropogenic climate change. Our results show that urgent change in how countries and territories protect and restore landscape-scale habitat is crucial as the international community gears up to embrace a new global biodiversity framework post-2020.

## Methods

**Protected areas.** We determined the structural connectivity of the global network of PAs by quantifying intact continuous pathways (areas largely devoid of high anthropogenic pressures) between PAs. Data on PA location and boundary were obtained from the May 2019 World Database of Protected Areas (WDPA)[70]. We only considered PAs that had a land area of at least 10 km$^2$. As China removed most of its PAs from the public May 2019 WDPA version, we used the April 2018 WDPA for China only, which contained the full set of Chinese PAs at the time. It is important to note that our statistics may differ from those reported by countries and territories due to methodologies and dataset differences used to measure terrestrial area of a country or territory.

**Measure of human pressure.** We used the latest global terrestrial human footprint (HFP) maps—a cumulative index of eight variables measuring human pressure on the global environment—to calculate the average human pressure between PAs[35]. While there are other human pressure maps[74–76], the HFP is a well-accepted dataset that provided a validation analysis using scored pressures from $3114 \times 1$ km$^2$ random sample plots. The root mean squared error for the 3114 validation plots was 0.125 on the normalized 0–1 scale, indicating an average error of approximately 13%. The Kappa statistic was 0.737, also indicating high concurrence between the HFP and the validation dataset. The HFP 2013 map uses the following variables: (1) the extent of built human environments, (2) population density, (3) electric infrastructure, (4) crop lands, (5) pasture lands, (6) roads, (7) railways, and (8) navigable waterways.

Navigable waterways such as rivers and lakes are included within HFP as they can act as conduits for people to access nature[35]. In the latest HFP (2013), rivers and lakes are included based on size and visually identified shipping traffic and shore side settlements. Venter et al. treated the great lakes of North America, Lake Nicaragua, Lake Titicaca, Lake Onega, Lake Peipus, Lake Balkash, Lake Issyk Kul, Lake Victoria, Lake Tanganyika and Lake Malawi, as they did navigable marine coasts (i.e. only considered coasts as navigable for 80 km either direction of signs of a human settlement, which were mapped as a night lights signal with a Digital Number (DN) > 6 within 4 km of the coast[35]. Rivers were included if their depth was >2 m and there were night-time lights (DN > = 6) within 4 km of their banks, or if contiguous with a navigable coast or large inland lake, and then for a distance of 80 km or until stream depth is too shallow for boats. To map rivers and their depth, Venter et al. used the hydrosheds (hydrological data and maps based on shuttle elevation derivatives at multiple scales) dataset on stream discharge, and the following formulae: stream width = 8.1× (discharge[m$^3$/s])0.58; and velocity = 4.0 × (discharge[m$^3$/s])0.6/(width[m]); and cross-sectional area = discharge/velocity; and depth = 1:5× area/width[35].

Each human pressure was scaled from 0–10, then weighted within that range according to estimates of their relative levels of human pressure following Sanderson et al.[77]. The resulting standardized pressures were then summed together to create the HFP maps for all non-Antarctic land areas[35].

Within the main manuscript, we defined intact land as any 1 km$^2$ pixel with a HFP value not higher than or equal to 4. Within this threshold, all areas with a HFP score higher than 4 are defined as nonintact. While previous analyses showed that a >4 score is a key threshold above which species extinction risk greatly increases[39], we recognized that there is no one true threshold, which impacts all species equally. Some species may require no human pressure to successfully disperse, while others might successfully navigate through more intensively modified landscapes. Therefore, we conducted our analyses for two additional HFP thresholds. The first used a HFP score <1 and the second incorporated all areas with a HFP < 10.

**Probability of connectivity.** The probability of connectivity network-based metric underlies the analysis performed[78], with adaptations to account for structural connectivity provided by intact lands. Probability of Connectivity (PC) is given by the following formula:

$$PC = \frac{\sum_{i=1}^{n} \sum_{j=1}^{n} a_i a_j p_{ij}^*}{A_L^2}, \tag{1}$$

where $n$ is the total number of PAs in the study area (i.e. landmass of continent, country or territory), $a_i$ and $a_j$ are the total area of PAs $i$ and $j$, $p^*_{ij}$ is the maximum product probability between PAs $i$ and $j$, and $A_L$ is the total area of the study area. The maximum product probability ($p^*_{ij}$) considers both direct connections (movement from $i$ to $j$ without using any other intermediate PA in the network) and indirect connections (movement from $i$ to $j$ facilitated by one or several other intermediate PAs acting as stepping stones). The maximum product probability ($p^*_{ij}$) is calculated through network analysis using the values of the direct dispersal probabilities between nodes ($p_{ij}$). In this analysis, $p_{ij} = 1$ when PA $i$ and $j$ are

connected (edge to edge) by a continuous pathway of intact land and $p_{ij} = 0$ if not. Both probabilities will be equal when the direct movement is the most favorable (probable) pathway between $i$ and $j$. $p^*_{ij}$ will be larger than $p_{ij}$ when intermediate stepping stones increase the structural connectivity between $i$ and $j$ beyond what is possible by using only the direct connection between them[78,79]. Therefore, two PAs may not be directly connected by intact lands (hence having $p_{ij} = 0$), but may be connected through an intermediate stepping-stone PAs, which would give $p^*_{ij} = 1$.

**Structural connectivity between protected areas.** The Probability of Connectivity (PC) metric accounts for both intra-PA ($i = j$) and inter-PA area ($i \neq j$) structural connectivity, which is, respectively, given by the intra-PA (PCintra) and inter-PA (PCinter) components of PC are

$$PC = PCintra + PCinter. \tag{2}$$

PCintra is calculated using the formula

$$PCintra = \frac{\sum_{i=1}^{n} \sum_{j=1, i=j}^{n} a_i a_j p_{ij}^*}{A_L^2} = \frac{\sum_{i=1}^{n} a_i^2}{A_L^2}. \tag{3}$$

While PCinter is mathematically defined as

$$PCinter = \frac{\sum_{i=1}^{n} \sum_{j=1, i \neq j}^{n} a_i a_j p_{ij}^*}{A_L^2}. \tag{4}$$

In this analysis, we investigated the connectivity between PAs (i.e. all PAs considered, regardless of how much intact land they contain) that is provided by intact land. For this reason, here the intra-node connectivity is removed and we focus only the inter-node (inter-PA) connectivity (PCinter) for both country/territory and continent level analyses. PCinter is defined as the probability that two points randomly located in two different PAs within the study area (therefore considering only the cases where $i \neq j$) are connected to each other via intact habitat. We calculated PCinter using two scenarios: PCinter_intact and PCinter_all. PCinter_intact is the value when considering that only the intact lands provide structural connectivity between PAs. PCinter_all is the value when any land (all land, intact, or not) provides structural connectivity between PAs, (i.e. considering that two PAs are connected when they are located in the same landmass or island). This analysis provided us with the maximum terrestrial PA structural connectivity that could be theoretically achieved in a country/territory or continent if all of its land was intact. In both scenarios, an 8-neighbouhood rule between land cells was used when defining the continuity of land (using the 1 km$^2$ resolution of the HFP layer).

**Structural connectivity provided by intact lands: ConnIntact.** We combined PCinter_intact and PCinter_all, as defined above, to obtain ConnIntact, which quantifies the percentage of the PA system that is connected through intact pathways. It is calculated using the following ratio:

$$ConnIntact = 100 \frac{PCinter\_intact}{PCinter\_all}. \tag{5}$$

Which, given the equation for PCinter above, can be expressed as:

$$ConnIntact = 100 \frac{\sum_{i=1}^{n} \sum_{j=1, i \neq j}^{n} a_i a_j p_{INTACT_{ij}}^*}{\sum_{i=1}^{n} \sum_{j=1, i \neq j}^{n} a_i a_j p_{ALL_{ij}}^*}, \tag{6}$$

where $p_{INTACT}$ refers to the maximum product probabilities when only the intact lands provide structural connectivity between PAs, $p_{ALL}$ refers to the maximum product probabilities when all land would be intact and hence would provide the highest possible structural connectivity between PAs, $n$ is the total number of PAs in the study area (e.g. a country, territory or continent), and $a_i$ and $a_j$ are the total area of PAs $i$ and $j$. To calculate connectivity, we consider all possible land between two PAs if they are located in the same landmass or island (i.e. if there is a continuous land pathway between the PAs). An 8-neighbourhood rule between land cells is used when defining the continuity of land. This analysis provides insight into how well connected the PAs would be if all land was intact (i.e. the proportional connection based on the maximum terrestrial PA connectivity that could be theoretically achieved in a country, territory, or continent). It is important to note that not all the PAs in a given country, territory, or continent will be connected if they are located in different landmasses or islands, but calculated as an aggregation of the results at the country or territory level. ConnIntact provides the percentage of the PA network that is connected by intact lands. This metric is expressed as a percentage of the total area under protection (see Supplementary Figs. 1–3).

**Theoretical examples.** The ConnIntact metric assumes that all PAs have the same area and that $n$ is the number of PAs in a hypothetic country. In addition, we define $t$ is the proportion of PAs that are located within the intact land. Therefore, $t \cdot n$ is the number of PAs within intact land.

If all the intact land is located in a single and continuous intact patch (so that all PAs within intact land are connected to each other), then $t^*n$ ($t^*n - 1$) is the number of PA pairs that are connected (both directions) by intact land. The maximum number of PA pairs that would be connected (both directions) if all the

land within the study area was intact would be $n \cdot (n-1)$. As ConnIntact is expressed as a percentage, the value of ConnIntact in this case is equal to $100 \cdot t \cdot n \cdot (t \cdot n - 1)/n \cdot (n-1)$ (a particular and simplified case of Eq. (5) in the main text).

In the Supplementary Fig. 1 example, $n = 20$; therefore, the maximum possible number of connections (pairs of PAs connected) is 380. Example 1a illustrates when the entire country is covered by one continuous patch of intact land, all PAs pairs are connected by intact land and ConnIntact is 100% (Supplementary Fig. 1a). This means that the 380 potential connections among PAs are all possible through intact land. Example 1b illustrates when we divide the country in two patches of intact land disconnected by nonintact land (Supplementary Fig. 1b) the protected areas on the right side become disconnected to the ones in the left side. However, inside each patch, PAs are still connected to each other by intact land. We can consider this case as the combination of two sets of PAs, $n_1 = 10$ and $n_2 = 10$, within which all PAs are connected. The number of PA pairs connected within each of the two intact land patches is therefore 90, which gives a total of 180 connections for the two intact land patches. This gives a value for ConnIntact of 47.4%, since 180 of the 380 potential connections are facilitated by intact pathways. Example 1c shows only ten of the PAs are located within a single patch of intact land, while the other 10 PAs are found in nonintact land (as in Supplementary Fig. 1c), then there are only 90 pairs of PAs connected via intact land and hence ConnIntact = 23.7%.

Within our analysis, connectivity is calculated from edge to edge and considers both direct and indirect connections. This means that two PAs are considered connected if they have a direct connection (a single patch of intact land connecting them), as is the case of all PAs in Supplementary Fig. 2b and 2c, but also if they have an indirect connection between them, facilitated by intermediate stepping-stone PAs. The latter is the case of the PAs labelled as X and Z in Supplementary Fig. 2a. PA X and PA Z are not directly connected (it is not possible to move from X to Z through a single continuous pathway of intact land). It is, however, possible to move from X to Y (edge to edge) through a continuous intact land patch, and from Y to Z (edge to edge) through another continuous intact land patch. Therefore, X is connected to Z as quantified by the connectivity analyses and ConnIntact metric here considered. In Supplementary Fig. 2, all the PAs are, in each of these three cases, connected by intact lands and have therefore the same value of the connectivity metric here considered (ConnIntact attains the maximum value of 100% in all these three cases).

We note that PCinter and ConnIntact can be used to evaluate the percentage of the PA pairs that are connected by intact lands, but cannot be used to state in general which PA network is 'best' or 'best' designed. Because PCinter and ConnIntact only consider inter-PA connections, it would be theoretically possible to have lower PCinter and ConnIntact for PA systems that are not more 'poorly connected' than others in certain comparisons. This is illustrated in Supplementary Fig. 3a; the ConnIntact metric, which quantifies how many pairs of PAs are connected (under the simplified case, as this one, in which all PAs have the same area), is equal to zero because the two PAs are isolated (not connected by intact lands). In Supplementary Fig. 3b, the ConnIntact metric is close to 50% (47.4%) because many of the pairs of smaller PAs are 'locally' (within each of the individual intact land patches) connected to each other. This comparison illustrates that, because the ConnIntact metric does not consider the intra-PA connectivity but focuses only in the inter-PA connectivity, it cannot be used to make a judgement about which of the PA systems is best. While ConnIntact is higher in Supplementary Fig. 3b than in Supplementary Fig. 3a, there are no reasons to think that, in general, Supplementary Fig. 3b can be regarded as a better PA system than Supplementary Fig. 3a.

**Three conditions analysis**. We converted the three conditions dataset to a raster, and snapped to the same resolution (1 km$^2$) and projection (Mollweide) as the HFP dataset. We then used the tabulate area tool in ArcGIS 10.6 to calculate the area of each condition per country or territory.

**Sensitivity analysis**. We tested the sensitivity of our results to the data obtained from the HFP score <4, using two additional HFP thresholds. With a HFP score <1, globally the proportion of connected PAs results in 8.2% rather than 9.7% for a HFP threshold of <4 (1.5% when considering the absolute difference in the proportion of connected PAs globally). When a HFP threshold of <10 is considered instead of <4, the proportion of the area under protection that is connected increases from 9.7% to 43.5% globally. This result does not alter our conclusions appreciably because the percentage of total land considered intact also varies in a similar fashion: 25% of all terrestrial land under <1 HFP threshold, 41.6% of all terrestrial land under <4 HFP threshold, and 74% of all terrestrial land under <10 HFP threshold.

Certain areas change considerably under different HFP thresholds. For example, the increase in the PA structural connectivity for HFP threshold <10 is particularly noticeable in Oceania (from 15.6 to 94.9%) and Russia (from 1.9 to 79%), and increases to more than 30% of PAs connected in Africa, Americas and Asia. This occurs because for this HFP threshold, most of the roads are no longer considered as a barrier, which can influence the movement of some particular species or group of species[17]. This suggests that the structural connectivity depends also on how species respond to the permeability of the landscape. Therefore, the

HFP threshold of <1 might be better to assess the structural connectivity for sensitive species to human activities such as the boreal woodland caribou, yet for species that can move through more human-modified landscapes, such as the American black bear, the HFP threshold of <10 might be more acceptable.

**Reporting summary**. Further information on research design is available in the Nature Research Reporting Summary linked to this article.

## Data availability
The HFP data are available for download from https://doi.org/10.5061/dryad.052q5, protected-area data (2019 version) are available for download from www.protectedplanet.net by request. All other data needed to evaluate the conclusions are present in the paper or Supplementary Information.

## Code availability
All relevant codes used in this work are available, upon request, from the corresponding author (M.W.).

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

## Acknowledgements

This research was supported by an Australian Government Research Training Program Scholarship and by the institutional activities of the Joint Research Centre of the European Commission. The work was funded by the NASA Biodiversity and Ecological Forecasting Program under the 2016 ECO4CAST solicitation through grant NNX17AG51G. N.A.D. was supported by the Fundación Bancaria 'la Caixa' Post-graduate Fellowship (LCF/BQ/AA16/11580053) and by the University of Queensland Research Training Scholarship.

## Author contributions

J.E.M.W. conceived the study, J.E.M.W., O.V., S.S., M.W., B.W., J.P.R.-D., J.A., and N.A. D. developed the idea and methods, S.S. and G.D. developed and globally applied the protected-area structural connectivity model, M.W. conducted the analysis of results and statistical evaluation, and M.W. led the writing of the manuscript with input from all authors.

## Competing interests

The authors declare no competing interests.
