## [Peer Review File · Nature Communications]

REVIEWER COMMENTS

Reviewer #1 (Remarks to the Author):

Summary

Protected areas (PAs) are cornerstones of conservation and the international community is making progress towards increasing the global coverage of terrestrial and marine protected areas. In order to be effective, PAs must be relatively intact and connected across a landscape/seascape. While we have a good understanding of how the global community is doing with regards to areal targets for PAs, few studies have quantified how connected PAs. In this contribution, the authors quantify the physical connectedness of global PAs and assess how these patterns vary across continents. The authors report that a small percentage of PAs can be considered connected, therefore, the global community may not be making good progress towards this conservation goal.

General comments:

1. I think the authors are filling a useful gap with a strong global analysis. There are many specific weaknesses inherent in the data used in these analyses but I believe the strengths of this global analysis outweigh the limitations. A global analysis such as this one will prove very useful in guiding research progress on this important conservation gap (i.e., connectedness of PAs).
2. The authors claim to study connectivity but I think the authors are studying physical connectedness. Connectivity is a process that can be species specific. I suspect the authors focus on "intact" areas as a proxy for connectivity (ie more intact is more likely to confer connectivity for species) but this is never made explicit in the paper. From a conservation perspective, the distinction between connectivity and connectedness is critically important and I worry that presenting the work as connectivity without any caveats will simply propagate the study and consideration of physical connectedness. Physical connectedness alone does not guarantee connectivity for a species. I recommend the authors think about this distinction in the paper and be explicit about connectivity vs connectedness.
3. I would appreciate some discussion (maybe in appendix?) of study limitations. I think this is important because this study may spark many similar studies and explicitly identifying some limitations will lead to stronger future studies. One aspect that deserves attention is where the analysis based on these data lies in the continuum of PA connectivity studies. On the one hand, these results could be considered conservative (i.e., high estimate of global PA connectedness) because the Human Footprint index, while a good data source, is still far from complete (i.e., it misses some critical human activities). On the other hand, these results could be considered extreme (i.e., low estimate of global PA connectedness) because there are PAs missing from the WDPA (e.g., private lands) and land-management policies can be very effective at creating defactor protected areas (think of OECM).
4. Many PAs have high human footprints inside their boundaries. What is the value of physical connectedness among PAs if a PAs is not intact within its own boundaries? I think this critical point deserves some discussion here. Related to this, why did the authors decide to remove the intra portion of the PA connectedness metric? This component has the potential to be useful to look at how connectedness among PAs varies as a function of connectedness/intactness within PAs.
5. The study is pitched as novel and the global analysis certainly is. But there has been a good deal of study at local or regional extents on the influence of areas surrounding PAs on PA effectiveness. These are not the same question as this study but I think some of these studies can provide a nice theoretical background to the current study. For example, DeFries et al. 2010 *Frontiers in Ecology & Environment*, Hansen & DeFries 2007 *Ecol Appl*, Hansen et al. 2011 *BioScience*.
6. How were land edges and waterbodies considered in the analysis? I struggled a bit to fully grasp the connectedness metrics. I recommend that authors create a figure with three worked (real) examples of landscapes and metric calculations to walk the reader through the process. I think this would be very valuable to encourage others to adopt the tools and approach and to help the reader interpret the findings.
7. Lines 157-160. I think this explanation is too simple. I think in many cases PAs establishment

falls under multiple jurisdictions within one country. For example, federal, provincial/state, municipal. Lack of connectedness of PAs established across jurisdictions may not occur simply because these jurisdictions do not work well together (i.e., they operate under different time frames, objectives, etc).

Shawn J. Leroux

Reviewer #2 (Remarks to the Author):

This paper provides a new method for measuring the connectivity of protected areas, a feature that is critical for protected areas to effectively achieve the conservation objectives for which they are designated. The authors do this by looking at the level of human pressure on the land between protected areas. The method used is a clear improvement on previous efforts to measure ecological connectivity and is therefore novel and a step forward in this area of research.

The manuscript is well written, clearly presented and easy to follow.

However, I am left wanting more and I don't feel that the authors do justice to their work. They need to go further and make recommendations on how this could be practically used in conservation, either by national governments wanting to expand their conservation area network, or by the global community who need an improved measure of connectivity for tracking the achievement of global goals. In line 227 they note the need for measures beyond area, a point which has been made many times, but they stop short of proposing a metric or a reporting mechanism that would achieve this. It is certainly a complex topic, this could take this paper from simply being a neat analysis of PA connectivity through to a policy relevant proposal for improving global conservation efforts.

The authors have used the Human Footprint index as their measure of intactness. It would be interesting to know if a similar pattern and result was seen by using a different measure of intactness, such as the Predicts model Biodiversity Intactness Index (Hudson et al. 2016), which takes a slightly different approach and looks at the biodiversity response to human pressures. Hudson, L. et al (2016) The database of the PREDICTS (Projecting Responses of Ecological Diversity In Changing Terrestrial Systems) project, Ecology & Evolution. <https://doi.org/10.1002/ece3.2579>
<https://www.bipindicators.net/indicators/biodiversity-intactness-index>

Likewise using a published and policy designed framework such as the Three Conditions for Nature (Locke et al 2019) instead of using a two level intact v's non-intact based on the HFP would provide more refined results linked to the national/regional condition. The authors correctly flag that a low connectivity score in regions such as Europe or Venezuela might not be a poor result (line 233, line 250). Adding an overlap of the three conditions may add some detail on the validity of the scores, help understand expected scores for a condition type, and hence support policy development in a more nuanced way.

Locke, H. et al (2019) Three Global Conditions for Biodiversity Conservation and Sustainable Use: an implementation framework. National Science Review DOI: 10.1093/nsr/nwz136.
<https://naturebeyond2020.com/3conditions/>

The manuscript focuses on ecological connectivity, but the authors should make reference to the fact that for protected areas connectivity to be successful there are also considerable governance, social and economic considerations. The IUCN have a number of guidance documents on connectivity where these points are expanded on.
<http://conservationcorridor.org/technical-guides/>

Line 211-213 Protected Areas connectivity and other effective area-based conservation measures

(OECM). The text here is not quite correct, although it is true that OECM will play an 'enormous role in keeping connectivity between PAs'. For the most part OECM are already in place on the ground and by definition they are effective. The key issue is that because countries and territories have only started identifying these areas they are not yet reported to the global databases managed by UNEP-WCMC. It is unlikely that the number will 'increase rapidly', rather the identification and reporting of the areas will start in earnest. Likewise, as they are already in place, it is more a question of the areas getting the recognition and support they need to continue conserving biodiversity, rather than them being 'planned and implemented'.

Finally – it is a great pity that the authors didn't tackle the marine environment. The lack of a good marine connectivity indicator is consistently flagged as a major gap in the global indicator set.

In summary, I think this is a very useful paper, but it needs an expansion of the analysis to make it more policy relevant and some bolder recommendations to ensure it can be used constructively in policy development. The paper is very timely, as ecological connectivity may well become an overarching goal the post2020 biodiversity framework to be agreed in 2021, and the indicators to track progress are currently under discussion.

Other points:

Figure 2a. This is a nice graph, but is quite hard to read and draw any conclusions from. Would it help to do the countries in size order by region? It appears that it is primarily small island states that are 'connected' (although I do note in Line 156 it says there is no significant relationship). It is useful to see the regional breakdown, but I wonder if there is a clearer way to present this.

Line 152. CBD not CDB.

Line 528, Citation for the WDPA is incorrect. Please see <https://www.protectedplanet.net/c/terms-and-conditions>. It should take the form: UNEP-WCMC and IUCN (year), Protected Planet: The World Database on Protected Areas (WDPA) [On-line], [insert month/year of the version downloaded], Cambridge, UK: UNEP-WCMC and IUCN. Available at: www.protectedplanet.net.

Reviewer #3 (Remarks to the Author):

Review of Ward et al. "Just ten percent of protected area network can be considered connected"

This paper evaluated connectivity between the world's protected areas and found that few of them can be considered connected. Protected areas are considered connected if land between them is "intact" (i.e., human footprint <4). The authors use their results to call for elevating the importance of connectivity in international conservation goals. They also acknowledge the importance of restoring land between protected areas to facilitate structural connectivity.

I highly recommend that the methods be made clearer. The main body of the paper and in the supplemental material, the authors make it seem as if they used a simple rule whereby protected areas with a contiguous swath of land where the human footprint <4 would be considered connected. If lands between protected areas did not include a continuous path where the human footprint <4 then the protected areas were considered not connected. However, the method focusing on probability of connectivity includes: "The maximum product probability (p^*_{ij}) is calculated through network analysis using the values of the direct dispersal probabilities between nodes (p_{ij})." This p^*_{ij} is key to the PC metric, but the sentence cited here does not provide enough information to understand how this was calculated. Are the "direct dispersal probabilities" either a 1 or 0 depending on whether a contiguous swath of land connected two protected areas or not? Was there some kind of dispersal distance where probability decays with distance from protected

area edge? I think it is important that these methods be made much clearer.

Was there a maximum distance that was considered when evaluating probability of connectivity between protected areas? If not how did you deal with Alaska when evaluating connectivity of the United States?

I appreciate the sensitivity analysis. It is clear and easy to follow. I do wonder though why the authors didn't use a resistance-based approach to evaluate connectivity between protected areas, rather than a hard classification rule. Belote et al. 2016 PLOS ONE uses a resistance-based least cost corridor approach to evaluate connectivity between protected areas, and Belote and Wilson 2020 Conservation Science and Practice used a cost-weighted distance approach to evaluate connectivity of the protected areas to the surrounding landscapes.

In formula 1 does a_j need to be defined in the text? I think I picked up that it is the area of the j th protected area, but I think it would be helpful to just spell this out as you did with a_i .

I didn't understand the importance of intra-PA connectivity in this paper. Does PC_{intra} in this case end up representing the proportion of area protected within a country?

Line 60. "disregarding the condition of the wider landscapes". See Belote and Wilson 2020 Conservation Science and Practice (DOI: 10.1111/csp2.196). In this paper, we acknowledge the importance of the wider landscape and used a cost-weighted distance approach using the human footprint as resistance to identify the lands relatively well connected to protected areas.

Line 243. I suggest changing "over-represented" to "well-represented".

Line 252. This is a really interesting and important point (corridors may be missed because of the 1-km resolution). It also made me think about whether protected areas that are connected only by 1 grid cell where the human footprint <4 would be considered connected. This also made me think of Beier's rule of thumb (2018 Con Bio) that suggest a 2-km wide corridor is a reasonable target for corridors.

Travis Belote
The Wilderness Society
Bozeman, MT USA

RESPONSE TO REVIEWERS

REVIEWER 1 COMMENTS

Comment:

Protected areas (PAs) are cornerstones of conservation and the international community is making progress towards increasing the global coverage of terrestrial and marine protected areas. In order to be effective, PAs must be relatively intact and connected across a landscape/seascape. While we have a good understanding of how the global community is doing with regards to areal targets for PAs, few studies have quantified how connected PAs. In this contribution, the authors quantify the physical connectedness of global PAs and assess how these patterns vary across continents. The authors report that a small percentage of PAs can be considered connected, therefore, the global community may not be making good progress towards this conservation goal.

1. I think the authors are filling a useful gap with a strong global analysis. There are many specific weaknesses inherent in the data used in these analyses but I believe the strengths of this global analysis outweigh the limitations. A global analysis such as this one will prove very useful in guiding research progress on this important conservation gap (i.e., connectedness of PAs).

Response:

We thank the reviewer for this positive review.

Comment:

2. The authors claim to study connectivity but I think the authors are studying physical connectedness. Connectivity is a process that can be species specific. I suspect the authors focus on “intact” areas as a proxy for connectivity (ie more intact is more likely to confer connectivity for species) but this is never made explicit in the paper. From a conservation perspective, the distinction between connectivity and connectedness is critically important and I worry that presenting the work as connectivity without any caveats will simply propagate the study and consideration of physical connectedness. Physical connectedness alone does not guarantee connectivity for a species. I recommend the authors think about this distinction in the paper and be explicit about connectivity vs connectedness.

Response:

We thank the reviewer for raising this very valid point. We agree that terminology around ‘connectivity’ could be an issue that does not need to detract from the analysis and have thoroughly edited the manuscript to clarify that in fact, we are measuring physical connectedness.

We have also changed the title to reflect this change: ‘*Just ten percent of the global terrestrial protected area network is structurally connected via intact land*’. This gets to the core issue around the term connectivity and connected that the reviewer raised.

We have also made this distinction in line 754 by stating: “We also note that structural connectivity is not the same as ecological connectivity, which has many components including species-specific functional connectivity. Functional connectivity measures the processes by which sub-populations of species are connected into a demographic unit¹⁸, and can be evaluated through strict adjacency⁷⁰, threshold distances⁷¹, or resistance-weighted functions⁷².”

And again in Line 201, we state: “Here, we analyze the structural connectivity of the global terrestrial PA system using measures of both the probability that connectedness can be achieved and contiguity of intact land (i.e. areas largely devoid of high anthropogenic pressures that significantly alter natural habitat). We assume that species can move more freely between PAs through intact land²⁵. We determined the connectedness of the global network of PAs by quantifying intact continuous pathways between PAs.”

Comment:

3. I would appreciate some discussion (maybe in appendix?) of study limitations. I think this is important because this study may spark many similar studies and explicitly identifying some limitations will lead to stronger future studies. One aspect that deserves attention is where the analysis based on these data lies in the continuum of PA connectivity studies. On the one hand, these results could be considered conservative (i.e., high estimate of global PA connectedness) because the Human Footprint index, while a good data source, is still far from complete (i.e., it misses some critical human activities). On the other hand, these results could be considered extreme (i.e., low estimate of global PA connectedness) because there are PAs missing from the WDPA (e.g., private lands) and land-management policies can be very effective at creating de factor protected areas (think of OECM).

Response:

We agree these points strengthen the discussion and have now added two comprehensive paragraphs outlining the study limitations based on the reviewer’s concerns. Starting line 746: “We note that our analysis is subject to some caveats. While the HFP is one of the most widely used and comprehensive datasets on global human pressure^{28,39,67}, it does not capture all human activities³⁵. For example, the HFP does not include walls or fences, which impose significant restrictions on species migrations and movement in some places⁶⁸. Invasive species are also a major contributor to the degradation of entire ecosystems and reduce functional connectivity⁶⁹, yet they are not directly mapped by the HFP. As such, our results may present an over-estimation of connectivity. Second, our analyses do not include some PAs that are not included within the WDPA⁷⁰ and when these places are mapped and embedded in the WDPA, the analysis should be updated. We also note that structural connectivity is not the same as ecological connectivity, which has many components including species-specific functional connectivity. Functional connectivity measures the processes by which sub-populations of species are connected into a demographic unit¹⁹, and can be

evaluated through strict adjacency⁷¹, threshold distances⁷², or resistance-weighted functions⁷³. While resistance-based approaches have also been used to evaluate structural connectivity between PAs¹⁶ and connectivity of the PAs to the surrounding landscapes¹¹, we did not use a resistance-based approach here because we wanted to take advantage of the latest HFP and the evidence that has emerged around the thresholds chosen^{28,36,39}. There is an important research gap to fill the links between our measures of structural connectivity to these efforts to measure functional connectivity.

We also note in some cases, a low proportional connectedness score between PAs may be a reasonable outcome. This is likely the case in some parts of Europe, where small-scale efforts to create corridors between PAs may not be captured at the 1km² resolution of our analysis. In addition, areas of high pressure in the matrix between PAs may be acceptable for some species that have co-existed with human-modified landscapes for centuries. But as most species, especially those that are endangered, cannot persist in human-dominated landscapes^{25,36}, our results should provide a sobering assessment for many nations.”

Comment:

4. Many PAs have high human footprints inside their boundaries. What is the value of physical connectedness among PAs if a PAs is not intact within its own boundaries? I think this critical point deserves some discussion here. Related to this, why did the authors decide to remove the intra portion of the PA connectedness metric? This component has the potential to be useful to look at how connectedness among PAs varies as a function of connectedness/intactness within PAs.

Response:

We agree this is an important point. We were specifically interested in how the PA estate, as defined and reported by nations, are connected, not the condition of their internal PA estate, which has been done in other studies such as Jones, K. R. et al. One-third of global protected land is under intense human pressure. *Science* 360, 788–791 (2018). If we were to include an assessment of the intra portion of the PA estate, we would be asking a broader question of how structurally connected intact habitats are, that have been protected. This would be interesting only if we then compared all non-protected intact landscapes, but would mean a very different question is answered.

We do, however, now explore this point in the discussion starting at Line 568: “We argue that these types of restoration goals should be framed within a broader connectedness agenda and specifically planned to maximize the quality of the landscape matrix between those PAs, as well as degraded land inside PAs essential to biodiversity outcomes. While we did not consider the condition inside PAs, we know that approximately one third of land protected is under intense human pressure⁸. Restoration of degraded land inside protected areas or even less degraded natural ecosystems outside but near PAs presents some of the most cost-effective restoration opportunities⁵⁵, and

lowest potential for conflicts with other priorities (such as agriculture). In addition, it is essential to incorporate the cost of such restoration to other societal goals when planning for conservation action.”

Comment:

5. The study is pitched as novel and the global analysis certainly is. But there has been a good deal of study at local or regional extents on the influence of areas surrounding PAs on PA effectiveness. These are not the same question as this study but I think some of these studies can provide a nice theoretical background to the current study. For example, DeFries et al. 2010 *Frontiers in Ecology & Environment*, Hansen & DeFries 2007 *Ecol Appl*, Hansen et al. 2011 *BioScience*.

Response:

We thank the reviewer for this suggestion and agree it is important to provide the most comprehensive background possible. As such, we have now cited six additional regional-scale papers in the introduction so as to make the important point the reviewer is suggesting. They include:

DeFries et al. 2010 *Frontiers in Ecology & Environment*

McGuire et al. 2016 *Proceedings National Academy of Sciences*

Hansen & DeFries 2007 *Ecol Appl*

Hansen et al. 2011 *BioScience*

Belote et al. 2016 *PLOS ONE*

Belote and Wilson 2020 in *Conservation Science and Practice*

Comment:

6. How were land edges and waterbodies considered in the analysis? I struggled a bit to fully grasp the connectedness metrics. I recommend that authors create a figure with three worked (real) examples of landscapes and metric calculations to walk the reader through the process. I think this would be very valuable to encourage others to adopt the tools and approach and to help the reader interpret the findings.

Response:

We agree with the reviewer about having worked examples and have now incorporated three methods figures with worked examples (see Supplementary information).

The *ConnIntact* metric calculated the maximum terrestrial PA connectivity that could be theoretically achieved in one landmass. Then using intactness, we measure what has actually been achieved. We have added to Line 494 to make this clear: “To calculate connectivity, we consider all possible land between two PAs if they are located in the same land mass or island (i.e. if there is a continuous land pathway between the PAs). An 8-neighbourhood rule between land cells is used when defining the continuity of land. This analysis provides insight into how well connected the PAs would be if all land was intact, (i.e. the proportional connection based on the maximum terrestrial PA connectivity that could be theoretically achieved in a country or continent). It is important to note that not all the PAs in a given country or continent will be connected if

they are located in different landmasses or islands, but calculated as an aggregation of the results at the country level.”

Waterbodies were only considered in the HFP if they met the HFP criteria for ‘Navigable waterways’. We have added to line 918 to make this clear: “Navigable waterways such as rivers and lakes are included with HFP as they can act as conduits for people to access nature. In the latest HFP (2013), rivers and lakes are included based on size and visually identified shipping traffic and shore side settlements. Venter et al. treated the great lakes of North America, Lake Nicaragua, Lake Titicaca in South America, Lakes Onega and Peipus in Russia, Lakes Balkash and Issyk Kul in Kazakhstan, and Lakes Victoria, Tanganyika and Malawi in Africa, as they did navigable marine coasts (i.e. only considered coasts as navigable for 80km either direction of signs of a human settlement, which were mapped as a night lights signal with a Digital Number (DN) >6 within 4km of the coast.) Rivers were considered as navigable if their depth was greater than 2m and there were signs of nighttime lights (DN>=6) within 4km of their banks, or if contiguous with a navigable coast or large inland lake, and then for a distance of 80 km or until stream depth is likely to prevent boat traffic. To map rivers and their depth, Venter et al. used the hydrosheds (hydrological data and maps based on shuttle elevation derivatives at multiple scales) dataset on stream discharge, and the following formulae: stream width = $8.1 (\text{discharge}[\text{m}^3/\text{s}])^{0.6}$; and velocity = $4.0 (\text{discharge}[\text{m}^3/\text{s}])^{0.6} / (\text{width}[\text{m}])$; and cross-sectional area = $\text{discharge}/\text{velocity}$; and depth = $1.5 \times \text{area}/\text{width}$.”

Comment:

7. Lines 157-160. I think this explanation is too simple. I think in many cases PAs establishment falls under multiple jurisdictions within one country. For example, federal, provincial/state, municipal. Lack of connectedness of PAs established across jurisdictions may not occur simply because these jurisdictions do not work well together (i.e., they operate under different time frames, objectives, etc).

Response:

Agreed and we thank the reviewer for this very valid point. We have now amended line 416 to state: “This indicates that structural connectivity may not be considered when nations are adding to their PA estates. For the more intact countries, this may be because they declare fewer but larger PAs, while more human-modified countries (such as those in Europe) declare many small PAs that are usually surrounded by a non-intact matrix. It may also be the result of PAs planning and management operating under multiple jurisdictions (e.g. federal, provincial/state, or municipal) within one country^{49,50}, with a lack of structural connectivity of PAs established occurring simply because of a lack of coordination between jurisdictions.”

REVIEWER 2 COMMENTS

Comment:

This paper provides a new method for measuring the connectivity of protected areas, a feature that is critical for protected areas to effectively achieve the conservation objectives for which they are designated. The authors do this by looking at the level of human pressure on the land between protected areas. The method used is a clear improvement on previous efforts to measure ecological connectivity and is therefore novel and a step forward in this area of research.

Response:

We thank the reviewer for these positive and insightful comments.

Comment:

The manuscript is well written, clearly presented and easy to follow. However, I am left wanting more and I don't feel that the authors do justice to their work. They need to go further and make recommendations on how this could be practically used in conservation, either by national governments wanting to expand their conservation area network, or by the global community who need an improved measure of connectivity for tracking the achievement of global goals. In line 227 they note the need for measures beyond area, a point which has been made many times, but they stop short of proposing a metric or a reporting mechanism that would achieve this. It is certainly a complex topic, this could take this paper from simply being a neat analysis of PA connectivity through to a policy relevant proposal for improving global conservation efforts.

Response:

We thank the reviewer for this extremely positive feedback. We agree more is needed around recommendations and have now added a paragraph, starting from Line 618, outlining how this framework could be used in combination with other published frameworks, which detail results and subsequent actions in a more nuanced way: “The method we use here is easily replicable. As such, it could provide a metric that governments can use, and report on, when creating new PAs that ensures this expansion results in a more connected PA network. The metric can also be used by the global community to measure, track, and implement global connectivity goals. For example, this effort to measure structural connectivity between PAs can be easily integrated with the policy implementation framework called the ‘Three Conditions for Nature’, that has been proposed to the CBD^{65,66}. This framework identifies three broad conditions of terrestrial areas based on land-use drivers and pressures. The three conditions include cities and farms (~18% of Earth), shared lands (~56% of Earth), and large wild areas (~26% of Earth)⁶⁵. Unsurprisingly, when structural connectivity between PAs is assessed under the Three Conditions for Nature framework, the majority of countries maintaining high proportions of connected PAs predominately also had high levels of large wild areas (Supplementary Table 4). Similarly, when countries have minimal connected PAs, the proportion of land within cities and farms and shared lands is predominately high. These areas require a very different set of actions varying

from restoration to retention of ecological integrity. By including these types of structural connectivity assessments within implementation frameworks like the Three Conditions for Nature, it is possible for decision makers to generate the different policy targets (e.g. more restoration in shared landscapes and more protection activities in wild landscapes) around connectivity that may lead to better outcomes.”

Comment:

The authors have used the Human Footprint index as their measure of intactness. It would be interesting to know if a similar pattern and result was seen by using a different measure of intactness, such as the Predicts model Biodiversity Intactness Index (Hudson et al. 2016), which takes a slightly different approach and looks at the biodiversity response to human pressures.

Hudson, L. et al (2016) The database of the PREDICTS (Projecting Responses of Ecological Diversity In Changing Terrestrial Systems) project, Ecology & Evolution. <https://doi.org/10.1002/ece3.2579>
<https://www.bipindicators.net/indicators/biodiversity-intactness-index>

Response:

We thank the reviewer for this suggestion. The PREDICTS model is very different as it integrates biophysical data to measure how intact the biodiversity assemblage is in a particular location – not how intact the habitat is. This index links data on land use with expert assessments of how this impacts the population densities of taxonomic groups to estimate current population sizes relative to premodern times. As reviewer 1 pointed out, we are actually measuring structural connectivity (which we have now made more clear throughout our manuscript), rather than species-specific connectivity. Therefore, if used and compared, it would answer a fundamentally different question.

It also seems that some believe that the PREDICTS model may have severely underestimated the extent of land degradation. For example, Martin et al. (*‘The biodiversity intactness index may underestimate losses’*) suggests that the overall index for southern Africa is probably much lower than 84%. They argue that this discrepancy is in part an artefact of the coarseness of the land degradation data used to calculate the index, and that the use of ground-truthed studies of areas not generally regarded as exceptional in terms of their degradation status. These differences have substantial bearing on index scores.

We do acknowledge that we only used cumulative human pressure map, and there are other datasets and approaches that could be used. We have now amended the manuscript to reflect this decision, starting from Line 909: “While there are other human pressure maps^{62–64}, the human footprint is a well-accepted dataset that provided a validation analysis using scored pressures from 3114 × 1km² random sample plots. The root mean squared error for the 3114 validation plots was 0.125 on the normalized 0–1 scale, indicating an

average error of approximately 13%. The Kappa statistic was 0.737, also indicating good agreement between the HFP and the validation dataset.”

Comment:

Likewise using a published and policy designed framework such as the Three Conditions for Nature (Locke et al 2019) instead of using a two level intact v's non-intact based on the HFP would provide more refined results linked to the national/regional condition. The authors correctly flag that a low connectivity score in regions such as Europe or Venezuela might not be a poor result (line 233, line 250). Adding an overlap of the three conditions may add some detail on the validity of the scores, help understand expected scores for a condition type, and hence support policy development in a more nuanced way.

Locke, H. et al (2019) Three Global Conditions for Biodiversity Conservation and Sustainable Use: an implementation framework. National Science Review DOI:10.1093/nsr/nwz136. <https://naturebeyond2020.com/3conditions/>

Response:

We thank the reviewer for this insightful comment. We have now conducted this additional analysis using the three conditions per country and have included the full results within the supplementary data. We have also added detail to the discussion starting from Line 618: “The method we use here is easily replicable. As such, it could provide a metric that governments can use, and report on, when creating new PAs that ensures this expansion results in a more connected PA network. The metric can also be used by the global community to measure, track, and implement global connectivity goals. For example, this effort to measure structural connectivity between PAs can be easily integrated with the policy implementation framework called the ‘Three Conditions for Nature’, that has been proposed to the CBD^{65,66}. This framework identifies three broad conditions of terrestrial areas based on land-use drivers and pressures. The three conditions include cities and farms (~18% of Earth), shared lands (~56% of Earth), and large wild areas (~26% of Earth)⁶⁵. Unsurprisingly, when structural connectivity between PAs is assessed under the Three Conditions for Nature framework, the majority of countries maintaining high proportions of connected PAs predominately also had high levels of large wild areas (Supplementary Table 4). Similarly, when countries have minimal connected PAs, the proportion of land within cities and farms and shared lands is predominately high. These areas require a very different set of actions varying from restoration to retention of ecological integrity. By including these types of structural connectivity assessments within implementation frameworks like the Three Conditions for Nature, it is possible for decision makers to generate the different policy targets (e.g. more restoration in shared landscapes and more protection activities in wild landscapes) around connectivity that may lead to better outcomes.”

Comment:

The manuscript focuses on ecological connectivity, but the authors should make reference to the fact that for protected areas connectivity to be successful there are

also considerable governance, social and economic considerations. The IUCN have a number of guidance documents on connectivity where these points are expanded on.

<http://conservationcorridor.org/technical-guides/>

Response:

We agree. We have expanded the discussion on Line 485 to say: “For Earth’s remaining intact and connected areas to be retained, they must be formally recognized, socially accepted, , prioritized in spatial plans, economically viable, and then placed under secure management plans, so they can be protected from human impacts⁵¹. Taking a coordinated approach of all factors can greatly benefit long-term connectivity as it provides the opportunity to identify areas that are most at risk of alienation, acquire patches of key importance to maintain connectivity, and can provide economic, social, and cultural needs for people⁵².”

We would also like to thank the reviewer for pointing us to the IUCN technical guides and have now referenced many of these within the manuscript.

Comment:

Line 211-213 Protected Areas connectivity and other effective area-based conservation measures (OECM). The text here is not quite correct, although it is true that OECM will play an ‘enormous role in keeping connectivity between PAs’. For the most part OECM are already in place on the ground and by definition they are effective. The key issue is that because countries and territories have only started identifying these areas they are not yet reported to the global databases managed by UNEP-WCMC. It is unlikely that the number will ‘increase rapidly’, rather the identification and reporting of the areas will start in earnest. Likewise, as they are already in place, it is more a question of the areas getting the recognition and support they need to continue conserving biodiversity, rather than them being ‘planned and implemented’.

Response:

Agreed and thank the reviewer for pointing this out. We have now amended Line 558 to read: “The global extent of identification and reporting of OECMs is expected to increase rapidly over the coming years. As such, their use, if planned and implemented well, could play an enormous role in keeping connectivity between PAs.”

Comment:

Finally – it is a great pity that the authors didn’t tackle the marine environment. The lack of a good marine connectivity indicator is consistently flagged as a major gap in the global indicator set.

Response:

Yes, we agree. We thank the reviewer for this great point. Unfortunately, it was outside of the scope for this paper but something we are very interested in pursuing in the future. The issue with marine connectivity is the principles of landscape ecology do not readily apply and while there has been wonderful advances in marine human footprint mapping, more fundamental science is needed to understand what connectivity means within these complex seascapes.

Comment:

In summary, I think this is a very useful paper, but it needs an expansion of the analysis to make it more policy relevant and some bolder recommendations to ensure it can be used constructively in policy development. The paper is very timely, as ecological connectivity may well become an overarching goal the post2020 biodiversity framework to be agreed in 2021, and the indicators to track progress are currently under discussion.

Response:

We thank the reviewer for this extremely positive feedback. We have now addressed all the comments, which has greatly improved the paper.

Comment:

Figure 2a. This is a nice graph, but is quite hard to read and draw any conclusions from. Would it help to do the countries in size order by region? It appears that it is primarily small island states that are 'connected' (although I do note in Line 156 it says there is no significant relationship). It is useful to see the regional breakdown, but I wonder if there is a clearer way to present this.

Response:

Agreed. We have now re-ordered the countries by size per region.

Comment:

Line 152. CBD not CDB.

Response:

Done

Comment:

Line 528, Citation for the WDPA is incorrect. Please see <https://www.protectedplanet.net/c/terms-and-conditions>. It should take the form: UNEP-WCMC and IUCN (year), Protected Planet: The World Database on Protected Areas (WDPA) [On-line], [insert month/year of the version downloaded], Cambridge, UK: UNEP-WCMC and IUCN. Available at: www.protectedplanet.net.

Response:

Done.

REVIEWER 3 COMMENTS

Comment:

Review of Ward et al. “Just ten percent of protected area network can be considered connected”

This paper evaluated connectivity between the world’s protected areas and found that few of them can be considered connected. Protected areas are considered connected if land between them is “intact” (i.e., human footprint <4). The authors use their results to call for elevating the importance of connectivity in international conservation goals. They also acknowledge the importance of restoring land between protected areas to facilitate structural connectivity.

Comment:

I highly recommend that the methods be made clearer. The main body of the paper and in the supplemental material, the authors make it seem as if they used a simple rule whereby protected areas with a contiguous swath of land where the human footprint <4 would be considered connected. If lands between protected areas did not include a continuous path where the human footprint <4 then the protected areas were considered not connected. However, the method focusing on probability of connectivity includes: “The maximum product probability (p^*_{ij}) is calculated through network analysis using the values of the direct dispersal probabilities between nodes (p_{ij}).” This p^*_{ij} is key to the PC metric, but the sentence cited here does not provide enough information to understand how this was calculated. Are the “direct dispersal probabilities” either a 1 or 0 depending on whether a contiguous swath of land connected two protected areas or not? Was there some kind of dispersal distance where probability decays with distance from protected area edge? I think it is important that these methods be made much clearer.

Response:

We thank the reviewer for this very important point. We have now created three methods figures with worked examples of landscapes and metric calculations to walk the reader through the process (see Supplementary information).

Yes, the direct dispersal probabilities are either a 1 if a contiguous land is connecting two protected areas, and 0 if it is not connected.

There was no dispersal distance where probability decays with distance from protected area edge, which fits nicely now with our amendments around structural connectivity.

We have amended the manuscript to capture these clarifications, starting with Line 201 stating: “Here, we analyze the structural connectivity of the global terrestrial PA system using measures of both the probability that connectedness can be achieved and contiguity of intact land (i.e. areas largely devoid of high anthropogenic pressures that significantly alter natural habitat). We assume that species can move more freely between PAs through intact

land²⁵. We determined the connectedness of the global network of PAs by quantifying intact continuous pathways between PAs.”

Line 976 has now been made clearer by stating: “The maximum product probability (p^*_{ij}) is calculated through network analysis using the values of the direct dispersal probabilities between nodes (p_{ij}). In this analysis, $p_{ij} = 1$ when protected area i and j are connected (edge to edge) by a continuous pathway of intact land and $p_{ij} = 0$ if not. Both probabilities will be equal when the direct movement is the most favorable (probable) pathway between i and j . p^*_{ij} will be larger than p_{ij} when intermediate stepping stones increase the connectedness between i and j beyond what is possible by using only the direct connection between them^{52,53,54}. Therefore, two protected areas may not be directly connected by intact lands (hence having $p_{ij}=0$), but may be connected through an intermediate stepping-stone protected areas, which would give $p^*_{ij}=1$.”

We have also added to line 971 stating that “where n is the total number of protected areas (PAs) in the study area (i.e. landmass of continent or country), a_i and a_j are the total area of PA i and j , p^*_{ij} is the maximum product probability between protected areas i and j , and A_L is the total area of the study area.”

Comment:

Was there a maximum distance that was considered when evaluating probability of connectivity between protected areas? If not how did you deal with Alaska when evaluating connectivity of the United States?

Response:

There was no maximum distance that was considered when evaluating probability of connectivity between protected areas, which fits nicely now with our amendments around structural connectivity rather than species-specific connectivity.

To calculate connectivity, we consider all possible land between two PAs as long as they are located in the same land mass or island (there is a continuous land pathway between the PAs). An 8-neighbourhood rule between land cells is used when defining the continuity of land. This analysis provides insight into how well connected the PAs would be if all land was intact, (i.e. the proportional connection based on the maximum terrestrial PA connectivity that could be theoretically achieved in a country or continent). It is important to note that not all the PAs in a given country or continent will be connected if they are located in different landmasses or islands, but calculated as an aggregation of the results at the country level. We now make this clear by incorporating a methods figure to the supplementary information and by adding to line 1098: “This analysis provided us with the maximum terrestrial protected area connectedness that could be theoretically achieved in a country or continent if all of its land was intact. In both scenarios, an 8-neighbourhood rule between land cells was used when defining the continuity of land (using the 1 km² resolution of the human footprint layer).

So for Alaska, this means that we calculated the structural connectivity of Alaska's PA network, then calculated the structural connectivity for the remainder of the United States. We then aggregated the results to estimate the country level results.

Comment:

I appreciate the sensitivity analysis. It is clear and easy to follow. I do wonder though why the authors didn't use a resistance-based approach to evaluate connectivity between protected areas, rather than a hard classification rule. Belote et al. 2016 PLOS ONE uses a resistance-based least cost corridor approach to evaluate connectivity between protected areas, and Belote and Wilson 2020 Conservation Science and Practice used a cost-weighted distance approach to evaluate connectivity of the protected areas to the surrounding landscapes.

Response:

The reviewer makes an important point around resistance-based approaches. Following reviewer 1's recommendation, we have now thoroughly edited the manuscript to clarify that in fact, we are measuring structural connectivity rather than functional connectivity and we have clearly stated what we are measuring and why.

We did not use a resistance-based approach because we wanted to take advantage of the human footprint and the evidence that has emerged around the thresholds we picked. A resistance approach would be mean making many more assumptions about thresholds and given species-specific responses to different levels of connectivity add a layer of complexity that we do not feel is supported in the literature at the moment. This being said, we agree that future approaches should adopt resistance-based approaches and we state this in the discussion around future research.

We have added this to the discussion starting Line 756: "Functional connectivity measures the processes by which sub-populations of species are connected into a demographic unit¹⁹, and can be evaluated through strict adjacency⁷¹, threshold distances⁷², or resistance-weighted functions⁷³. While resistance-based approaches have also been used to evaluate structural connectivity between PAs¹⁶ and connectivity of the PAs to the surrounding landscapes¹¹, we did not use a resistance-based approach here because we wanted to take advantage of the latest HFP and the evidence that has emerged around the thresholds chosen^{28,36,39}. There is an important research gap to fill the links between our measures of structural connectivity to these efforts to measure functional connectivity."

Comment:

In formula 1 does a_j need to be defined in the text? I think I picked up that it is the area of the j th protected area, but I think it would be helpful to just spell this out as

you did with a_i .

Response:

We thank the reviewer for pointing this out. We have now added to line 971 stating that “where n is the total number of protected areas (PAs) in the study area (i.e. landmass of continent or country), a_i and a_j are the total area of PA i and j , p^*_{ij} is the maximum product probability between protected areas i and j , and A_L is the total area of the study area.”

Comment:

I didn't understand the importance of intra-PA connectivity in this paper. Does PC_{intra} in this case end up representing the proportion of area protected within a country?

Response:

We thank the reviewer for this question. In Saura et al. (2018), intra-PA accounts for the amount of protected land that is available within individual PAs. Here, we do not use PC_{intra} because we were specifically interested in how the PA estate, as defined and reported by nations, are connected, not the condition of their internal PA estate, which has been done in other studies such as Jones, K. R. et al. One-third of global protected land is under intense human pressure. *Science* 360, 788–791 (2018). If we were to include an assessment of intra portion of the PA estate, we would be asking a more broader question of how structural connected intact habitats are, that have been protected. This would be interesting only if we then compared all non-protected intact landscapes but would mean a very different question is answered.

But we now explore this point in the discussion starting at Line 568: “We argue that these types of restoration goals should be framed within a broader connectivity agenda and specifically planned to maximize the quality of the landscape matrix between PAs, as well as degraded land inside PAs essential to biodiversity outcomes. While we did not consider the condition inside PAs, we know that approximately one third of land protected is under intense human pressure⁹. Restoration of degraded land inside PAs or less degraded natural ecosystems outside but near PAs presents some of the most cost-effective restoration opportunities⁵⁹, and lowest potential for conflicts with other priorities (such as agriculture). In addition, it is essential to incorporate the cost of such restoration and conservation actions into other societal goals.”

We have also amended Line 1088 to state: “In this analysis, we investigated the connectivity between PAs (i.e. all PAs considered, regardless of how much intact land they contain) that is provided by intact land. For this reason, here the intra-node connectivity is removed and we focus only on the inter-node (inter-PA) connectivity for both country and continent level analysis.”

Comment:

Line 60. “disregarding the condition of the wider landscapes”. See Belote and Wilson 2020 Conservation Science and Practice (DOI: 10.1111/csp2.196). In this paper, we acknowledge the importance of the wider landscape and used a cost-weighted distance approach using the human footprint as resistance to identify the lands relatively well connected to protected areas.

Response:

We thank the reviewer for drawing our attention to this important article. We have now edited Line 124 to state: “Yet, to date, reporting has been almost completely blind to how well connected the expanding global PA estate is, with only substantive research conducted at country and region scales^{11–16}, or solely considering connectivity through protected land^{17,18}, disregarding the condition of the wider landscape context.”

Comment:

Line 243. I suggest changing “over-represented” to “well-represented”.

Response:

Done.

Comment:

Line 252. This is a really interesting and important point (corridors may be missed because of the 1-km resolution). It also made me think about whether protected areas that are connected only by 1 grid cell where the human footprint <4 would be considered connected. This also made me think of Beier’s rule of thumb (2018 Con Bio) that suggest a 2-km wide corridor is a reasonable target for corridors.

Response:

We thank the reviewer for this thoughtful comment. As reviewer 1 pointed out, our analysis is more focused on measuring structural connectivity rather than functional connectivity. We have now thoroughly edited the manuscript to clarify this point, including changing the title to ‘*Just ten percent of the global terrestrial protected area network is structurally connected via intact land*’. This gets to the core issue around functional connectivity vs. structural connectedness. We have also made this distinction in line 201, by stating, “Here, we analyze the structural connectivity of the global terrestrial PA system using measures of both the probability that connectedness can be achieved and the contiguity of intact land (i.e. areas largely devoid of high anthropogenic pressures that significantly alter natural habitat) to quantify the probability that connectivity can be achieved. In this analysis, we assume that most species can move more freely between PAs through intact land²⁸. We determined the structural connectivity of the global network of PAs by quantifying intact continuous pathways between PAs.”

REVIEWERS' COMMENTS:

Reviewer #1 (Remarks to the Author):

The authors have done an excellent job of addressing all reviewer concerns in this revised ms. I very much appreciate the new conceptual diagrams working through the metric (in SI), the additional details in the methods, and added precision on the use of connectivity vs structural connectedness and study limitations. Congratulations!

Shawn J. Leroux

Reviewer #2 (Remarks to the Author):

Thank you for the opportunity to review this manuscript again. It is much improved and I very much appreciate the efforts the authors have taken to being on board the reviewer comments, including a lot of additional analyses.

My only final suggestion (which I should have picked up previously) is that the authors have done the analysis on 'countries and territories'. Therefore the use of the term 'country' or 'nation' throughout is not accurate and should be replaced with 'countries and territories'. It is particularly obvious in Figure 2 and 3 where a number of territories are picked out (e.g Greenland and Svalbard).

Naomi Kingston

Reviewer #3 (Remarks to the Author):

Thank you for considering my comments. The new supplemental figures are extremely helpful! I think they will strengthen the clarity of the methods, results, and interpretation. Very nice.

I also appreciate the added clarity for the context of the methods. Thank you.

RESPONSE TO REVIEWERS

REVIEWER 1 COMMENTS

Comment:

The authors have done an excellent job of addressing all reviewer concerns in this revised ms. I very much appreciate the new conceptual diagrams working through the metric (in SI), the additional details in the methods, and added precision on the use of connectivity vs structural connectedness and study limitations. Congratulations!

Shawn J. Leroux

Response:

We thank Dr. Leroux for his excellent and detailed review. The manuscript has been immensely improved from his input, particularly around terminology and clarity of methods.

REVIEWER 2 COMMENTS

Comment:

Thank you for the opportunity to review this manuscript again. It is much improved and I very much appreciate the efforts the authors have taken to being on board the reviewer comments, including a lot of additional analyses.

My only final suggestion (which I should have picked up previously) is that the authors have done the analysis on 'countries and territories'. Therefore the use of the term 'country' or 'nation' throughout is not accurate and should be replaced with 'countries and territories'. It is particularly obvious in Figure 2 and 3 where a number of territories are picked out (e.g Greenland and Svalbard).

Naomi Kingston

Response:

We thank Dr. Kingston for her wonderful suggestions, particularly around the three conditions. We believe this has greatly improved the paper.

We would also like to thank Dr. Kingston for her point on the terminology. We have now changed all text from 'nations' or 'countries' to 'countries and territories'.

REVIEWER 3 COMMENTS

Comment:

Thank you for considering my comments. The new supplemental figures are extremely helpful! I think they will strengthen the clarity of the methods, results, and interpretation. Very nice.

I also appreciate the added clarity for the context of the methods. Thank you.

Response:

8th June 2020

We thank Reviewer 3 for his suggestions. We agree that the methods are now much clearer with the added detail and figures, as suggested.